# Computational and Experimental Analysis of Gold Nanorods in Terms of Their Morphology: Spectral Absorption and Local Field Enhancement

**DOI:** 10.3390/nano11071696

**Published:** 2021-06-28

**Authors:** Juan Manuel Núñez-Leyva, Eleazar Samuel Kolosovas-Machuca, John Sánchez, Edgar Guevara, Alexander Cuadrado, Javier Alda, Francisco Javier González

**Affiliations:** 1Coordinación para la Innovación y Aplicación de la Ciencia y la Tecnología, Universidad Autónoma de San Luis Potosí, 550 Sierra Leona Ave, San Luis Potosí 78210, Mexico; A205030@alumnos.uaslp.mx (J.M.N.-L.); samuel.kolosovas@uaslp.mx (E.S.K.-M.); sanchezjohneder@gmail.com (J.S.); edgar.guevara@uaslp.mx (E.G.); javier.gonzalez@uaslp.mx (F.J.G.); 2Doctorado Institucional en Ingeniería y Ciencia de Materiales (DICIM-UASLP), Universidad Autónoma de San Luis Potosí, 550 Sierra Leona Ave, San Luis Potosí 78210, Mexico; 3Escuela Superior de Ciencias Experimentales y Tecnología, Universidad Rey Juan Carlos, C/ Tulipán s/n, Móstoles, 28933 Madrid, Spain; alexander.cuadrado@urjc.es; 4Applied Optics Complutense Group, Faculty of Optics and Optometry, University Complutense of Madrid, 118 Arcos de Jalón Ave, 28037 Madrid, Spain

**Keywords:** computational electromagnetism, gold nanorods, Raman spectroscopy

## Abstract

A nanoparticle’s shape and size determine its optical properties. Nanorods are nanoparticles that have double absorption bands associated to surface plasmon oscillations along their two main axes. In this work, we analize the optical response of gold nanorods with numerical simulations and spectral absorption measurements to evaluate their local field enhancement—which is key for surface-enhanced Raman spectroscopic (SERS) applications. Our experimental results are in good agreement with finite element method (FEM) simulations for the spectral optical absorption of the nanoparticles. We also observed a strong dependence of the optical properties of gold nanorods on their geometrical dimension and shape. Our numerical simulations helped us reveal the importance of the nanorods’ morphology generated during the synthesis stage in the evaluation of absorption and local field enhancement. The application of these gold nanorods in surface-enhancement Raman spectroscopy is analyzed numerically, and results in a $5.8\times 10^{4}$ amplification factor when comparing the values obtained for the nanorod deposited on a dielectric substrate compared to the nanorod immersed in water.

## 1. Introduction

The field of nanophotonics—interaction of light and matter at the subwavelength scale—has provided new strategies for optical sensing, energy harvesting, and the development of new optical devices [1,2,3,4]. Resonant nanostructures and nanoparticles are capable of enhancing the optical response and improving the performance of photonic devices [5,6,7,8,9,10,11,12].

A significant number of applications in nanophotonics depend on the nanoparticles’ geometry, and material composition [13,14,15,16,17]. Metallic nanoparticles have attracted great interest in biomedical applications due to their optical, electrical, and magnetic properties as a function of their size and morphology which can be tuned during the synthesis process [18,19,20,21]. A special case is gold nanoparticles, that exhibit unique properties used in biomedical applications [22,23,24,25,26,27,28].

We have chosen to work with nanorods because of their double absorption bands exhibited by surface plasmons associated with oscillations along the two main axes of the nanorod’s geometry: longitudinal (with rotational simmetry) and tranverse. The two characteristic lengthts of the nanorod’s geometry are intrinsically related to the excitation of surface plasmons at two different wavelengths. The presence of these two modes gives them an advantage over other nanoparticle shapes. This advantage is related to the capability of exciting two resonances. This characteristic can help amplify the Raman effect—strong field enhancement that increase the optical response—for two wavevelengths with a single nanoparticle [29,30,31,32,33].

Different approaches have been developed to model nanorods. Analytical solutions based on Mie theory applied to spheres have been expanded to obtain various optical parameters of non-spherical particles and arbitrary shapes; among these, a practical approach uses a T-matrix to apply the radiative transfer theory to light scattering by particles without rotational symmetry [34,35,36,37,38,39,40,41]. From a computational electromagnetism point of view, finite element methods (FEM) have evaluated the potential of nanorods in surface-enhanced Raman spectroscopy (SERS) [42,43,44,45]. Moreover, the finite difference time domain (FDTD) method has been used to model the localized surface plasmon resonance of nanorods [46,47]; also, the discrete dipole aproximation (DDA) is used to investigate the optical properties of plasmonic nanorods arrays [48,49,50,51,52,53]. These numerical methods can be used to predict the actual optical behavior of nanorods. However, the final results must be compared to actual optical measurements.

Once the simulation is deemed reliable, we can analyze the influence of geometrical and material parameters in the system’s overall behavior. Also, we can determine the extent of the electromagnetic interaction at the nanoscale, providing values of parameters that could require a complex ica measurement set-up (e.g., the field-enhancement factor).

In this work, we investigate the interaction of gold nanorods (AuNRs) with electromagnetic radiation in terms of wavelength, state of polarization, geometrical parameters, and relative orientation. We have evaluated the spectral optical absorption of AuNRs in terms of their size and orientation with respect to the incoming electric field, performing a novel computational model that considers the experimentally obtained size distribution. The results derived from simulations are compared to experimental measurements. We have also analized, through a complete and concise electromagnetic simulation, the ability of the our synthesized nanorods to obtain Raman signal amplification factors, as reported in experimental investigations [54,55,56,57,58,59].

We have organized this paper as follows. In Section 2, we have described the synthesis method to obtain our AuNRs (Section 2.1), the main conditions to generate a reliable simulation (Section 2.2), and the system used to measure the spectral absorption of the fabricated AuNRs (Section 2.3). Section 3 shows the analysis of the morphology and size distribution of the AuNRs. These results are in good agreement with the measured optical absorption (Section 3.1). We evaluate the synthesized nanoparticle’s application to Raman spectroscopy experiments by analyzing the electric field enhancement (Section 3.2). Finally, the main conclusions of this paper are summarized in Section 4.

## 2. Materials and Methods

### 2.1. Synthesis and Morphological Analysis of Gold Nanorods

Metallic nanorods can be synthesized using a variety of methods [60,61,62,63,64,65]. In this work, we obtained AuNRs using the Nikoobakht method [66], with the suggestions reported in previous works [67,68,69].

Our synthesis method used the following reagents: auric tetrachloride (HAuCl${}_{4}$, M=339.79 g/mol), sodium borohydried (NaBH${}_{4}$, M=37.83 g/mol), hexadecyltrimethylammonium bromide (CTAB, M=364.4 g/mol), L-ascorbic acid (C${}_{6}$H${}_{8}$O${}_{6}$, M=176.12 g/mol), hydrochloric acid (HCl, M=36.46 g/mol), silver nitrate (AgNO${}_{3}$, M=169.87 g/mol). HAuCl${}_{4}$, NaBH${}_{4}$, CTAB and C${}_{6}$H${}_{8}$O${}_{6}$ were provided by Sigma Aldrich (St. Louis, MO, USA), and HCl and AgNO${}_{3}$ were from Fermont (Nuevo León, Mexico). The seed solution used 10 mL of CTAB at 0.1 M, 600 μL of NaBH${}_{4}$ at 0.01 M, and 150 μL of HAuCl${}_{4}$ at 0.01 M. The growth of the seed to generate nanorods used 10 mL of CTAB at 0.01 M, 500 μL of HAuCl${}_{4}$ at 0.01 M, 80 μL of AgNO${}_{3}$ at 0.01 M, and 200 μL of HCl at 1 M. Inmediately after mixing these compounds, we added 80 μL of ascorbic acid at 0.1 M. Once the preparation changes color, we added 12 μL of the seed solution to generate the gold nanorods. We have used AgNO${}_{3}$ as a reducing agent. In addition, we used HCl in the growth process as a modifier of the pH value of the initial seed solution. This lower pH promotes larger aspect ratio values of the generated nanoparticles, as reported in previous studies [70,71,72,73,74].

The resulting nanoparticles were characterized through Scanning Electron Microscopy (SEM) using an Inspect F50 FEG-SEM electron microscope. Figure 1a was taken after depositing a drop of the nanorods solution on a silicon wafer. We sonicated the colloidal solution and allowed it to dry on the substrate. The nanoparticles show a rod morphology with the presence of small agglomerates and the formation of spherical terminations caused by the surfactant agent CTAB. The SEM images were processed using ImageJ software to determine the geometrical dimensions of the nanoparticles.

### 2.2. Computational Analysis of the Electromagnetic Response

Several techniques and approaches allow understanding better the nanorods’ behavior when illuminated by optical radiation. This can be implemented by software packages, open-source or proprietary, that provide reliable results after setting the appropriate initial and boundary conditions. In this contribution, we have made our analysis using a commercial FEM package, Comsol Multiphysics, that allows a complete characterization of the variables of interest. Our previous experience with this tool and our results’ reliability have also helped to interpret the results [33,75,76,77].

Once the dimensions of the nanoparticles were measured, we included them within our FEM model. AuNRs were first simulated as prolate ellipsoids, where the minor axes (transverse) are equal ($a=b=d_{t}$), and the major axis (longitudinal) is $c=d_{l}$; with an aspect ratio h=c/a that is ∼3 (see Figure 2a). This first approach has been refined to resemble the actual nanorods’ morphology by adding two hemispheres to a cylindrical body (see Figure 2b).

Several studies extrapolate Mie’s theory to obtain characteristic parameters for non-spherical particles and arbitrary shapes [36,37,38,40,78]. One of these works was developed by R. Gans [39], where Mie theory is extended to rod-shaped structures, simulating these geometries as prolate ellipsoids and obtaining anaytical solutions for their electromagnetic interaction. This method is the simplest approach for a nanorod. Nevertheless, when this type of geometry is extrapolated to the actual shape of a real nanorod, discrepancies are found. The geometry should be refined: we consider a shape obtained from a cylinder terminated by two hemispherical caps because it is closer to the real geometry of the nanoparticle.

From a technical point of view, we have completed two rounds of simulations depending on the variable of interest (major and minor axis of the AuNRs). To calculate the spectral absorption cross-section, we placed the nanoparticle in a spherical calculation domain filled with water. The sphere is limited using a scattering boundary condition at the interface between the water sphere and a surrounding perfect matched layer. The incoming wavefront is modeled as a monochromatic plane wave with an electric field $E_{0}=1$ V/m linearly polarized. This polarization state was oriented to excite the nanorod’s plasmonic resonances. We have included the substrate within our calculation domain when evaluating the field enhancement generated by the nanorods. To properly account for it, we consider a rectangular symmetry, and used the total field/scattered field method [79,80,81,82]. This procedure requires calculating the electromagnetic fields in the structure excluding the scatterers, but including the appropriate perfect matched layers surrounding the calculation domain. This total field is used as the background field that interacts with the nanorod in a second analysis. In this case, the electric field is injected from a port located at the water side as a monochromatic linearly polarized plane wave having an amplitude $E_{0}=1$ V/m.

### 2.3. Measurement of the Spectral Absorption

The spectral optical transmission of the fabricated nanorods was measured using a PerkinElmer (Lambda 25 UV/VIS Spectrometer model) with a 200–1000 nm measurement range. We performed an initial calibration to subtract the baseline; then, we added a colloidal sample of AuNRs into the quartz cells and placed it into the instrument. The spectra show some noise at short wavelengths below our region of interest, between 200 and 400 nm. The spectral locations of the peaks were found as the local maximum of the measured spectra.

## 3. Results and Discussion

### 3.1. Absorption Cross-Section, $\sigma_{\mathrm{abs}}$, and Spectral Absorption

The synthesis method explained in Section 2.1 has produced a collection of AuNRs that have been dimensionally characterized. The histograms of the nanoparticles’ longitudinal and transversal sizes observed in Figure 1a provide $d_{l}=52.3\pm 9.6$ nm and $d_{t}=17.8\pm 6.1$ nm, respectively (see Figure 1b,c). From this morphological analysis we can conclude that our synthesis has been successful because the nanoparticles have a nanorod shape with an aspect ratio of around 3 ± 1.

Using the previously defined conditions (see Section 2.2), we have used Comsol Multiphysics to better understand the nanostructures’ electromagnetic behavior. First, we calculated the spectral absorption cross-section, $\sigma_{\mathrm{abs}}$, as a function of the dimensions of the nanorod: $d_{l}$ for the field-oriented along this direction (θ=0°), and $d_{t}$ for the field-oriented at θ=90°. Considering the metallic character of our nanorods, we can assume that absorption is much stronger than scattering, and the results for $\sigma_{\mathrm{abs}}$ describes well the spectral behavior of these nanoparticles (see Appendix A). This calculation was completed for both geometries (prolate ellipsoid and cylinder+hemispherical caps). In Figure 3, we show the value of $\sigma_{\mathrm{abs}}$ as a map in terms of the wavelength, λ, and the most relevant dimension for the resonance at the two orthogonal polarization states: θ=0° when the electric field, $\vec{E}$, is aligned along the major axis of the geometry (with dimension $d_{l}$), and θ=90° for the case of $\vec{E}$ parallel to the minor axis (with dimension $d_{t}$). We also used the observed length distribution of our nanorods (see Figure 1b,c) to obtain the weighted average of $\sigma_{\mathrm{abs}}$ using the fitted Gaussian distribution. These spectral dependencies are shown in the second column of Figure 3. We observe that nanorods behave almost the same for both geometries. The spectral peaks show the plasmonic resonances associated to each axis. In Table 1, we describe the resonant wavelengths for the each geometry and the spectral location of the experimental peaks in absorption.

From the results of the measurements, we extracted the spectral absorption of the sample. Figure 4 shows the experimental results along with the absorption calculated for the two geometries. The plots related to the numerical calculation were evaluated as the average of the two spectral responses for θ=0° and θ=90°, due to the contribution of plasmonic modes of AuNRs and the size distribution present in the sample. The spectral location of the maximum of the absorption is also included in Table 1 and corresponds with the two expected plasmonic resonances linked to the nanorods’ axes ($d_{l}$ and $d_{t}$). From Figure 4, we also evaluated the agreement between the numerical results and the measurements quantitatively. We used the correlation coefficient between the normalized absorption cross-section, ${\hat{\sigma }}_{\mathrm{abs}}\left(\lambda \right)$, and the normalized experimental absorption, $\hat{A}\left(\lambda \right)$, as:(1)$$c=\frac{\int {\hat{\sigma }}_{\mathrm{abs}}\left(\lambda \right)\hat{A}\left(\lambda \right)d\lambda }{\int {\hat{\sigma }}_{\mathrm{abs}}\left(\lambda \right)d\lambda \int \hat{A}\left(\lambda \right)d\lambda }.$$

The results of this calculation (see Table 2) show that the geometrical model with cylinders terminated with hemispherical caps agrees with the experimental results better than the prolate ellipsoid shape, particularly for the case of the strongest resonance at θ=0°. So, we will consider the cylinder+hemispherical caps in the subsequent analysis of the electric field enhancement. This geometrical configuration is also closer to the observed shape of the fabricated nanorods than the prolate ellipsoid (see Figure 1a).

### 3.2. Near Field Enhancement

Nanoparticles can amplify effects linked with the electric field’s modulus. This field is strongly enhance by the presence of a substrate. Applied to our case, we have compared the electric fields of AuNRs immersed in water with those obtained when AuNRs are deposited on a dielectric substrate. Figure 5a,b show the spectral dependence of the maximum of the electric field’s modulus, where the presence of the substrate strongly enhances this modulus. The maps (in logarithmic scale) in Figure 5c,d represent the electric field’s spatial distribution when it is aligned along the nanorod’s long axis (θ=0°) at resonance. For an input electric field with amplitude $E_{0}=1$ V/m, these maps also represent the spatial distribution of the field enhancement. The enhancement factor is $5.8\times 10^{4}$ at λ=728 nm when comparing the ratio of maximum values of spectra with and without substrate. This value is of the same order as that obtained experimentally in previous works [58,83,84]. The field enhancement is generated in a spatial region close to the hemispherical caps. Although constrained both spectrally and spatially, this strong field enhancement may boost the Raman signal, which would help to lower the limit of detection of substances or biomolecules. This application has been previously demonstrated in the detection of biological substances using gold nanoparticles with Raman spectroscopy [33,85,86,87]. Additionally, if the AuNRs are functionalized to trap specific biomolecules, the capabilities of these nanoparticles expand and could detect proteins, biomarkers, antigens, and other biological compounds.

To exploit the nanorods’ field enhancement capabilities, our synthesized AuNRs can be deposited on a dielectric substrate and become the active layer in a sensing device. It would generate a strong resonance readable optically both angularly and spectrally, similar to conventional plasmonic sensors.

## 4. Conclusions

In this contribution, we present the experimental results obtained from synthesized gold nanorods that can be used in surface-enhanced Raman spectroscopy. We have synthesized gold nanorods with a longitudinal and transversal size that excite plasmonic resonances in the visible range. The aspect ratio (length to width) is around 3 ± 1. We have modeled two types of nanorods geometries—prolate ellipsoids and cylinders terminated with two hemispherical caps—with a computational electromagnetism tool. First, we calculated the spectral absorption cross-section for both geometries as a function of the nanorods’ dimensions. From the size distribution of the synthesized nanorods (taken from SEM images), we obtained the expected absorption by weighting the calculated spectral cross-section with the measured size distribution. We compared them with the experimental absorption obtained in the lab. Although the agreement is quite good for both geometries, the cylinder terminated with hemispherical caps provides a larger correlation coefficient and it also resembles the observed gold nanorods’ shape more faithfully.

After checking that the calculated absorption cross section and the experimentally measured absorption coincide in good agreement, we calculated the field enhancement generated by the nanorods. As expected, this field enhancement is strongly amplified when locating the gold nanorods on a dielectric substrate. The amplification factor is $∼5.7\times 10^{4}$ for the ratio of electric field with and without substrate. This value should generate an amplification in the Raman signal at the resonant wavelenth located spatially at some very specific regions around the hemispherical caps. Our gold nanorods—properly functionalized to trap the biomolecules of interest— are an excellent candidate for surface-enhanced Raman spectroscopy; specifically in biomedical, industrial and defense applications such as detection of biomarkers for cancer, delivery drugs, hazardous compounds, explosives, etc.

We can conclude that the analysis method used to obtain the correlation between simulated and experimental optical absorption obtains excellent reliability after considering the measured size distribution and dependence, and the real shape of the nanorods. We can also conclude that our gold nanorods have become a feasible candidate to be incorporated in an optical sensor able to detect the presence of biomolecules and other compounds through Raman spectroscopy. Our analysis has been supported by a reliable simulation method and a model that predicts the field enhancement characteristics of the fabricated nanoparticles.

## Figures and Tables

**Figure 1 nanomaterials-11-01696-f001:**
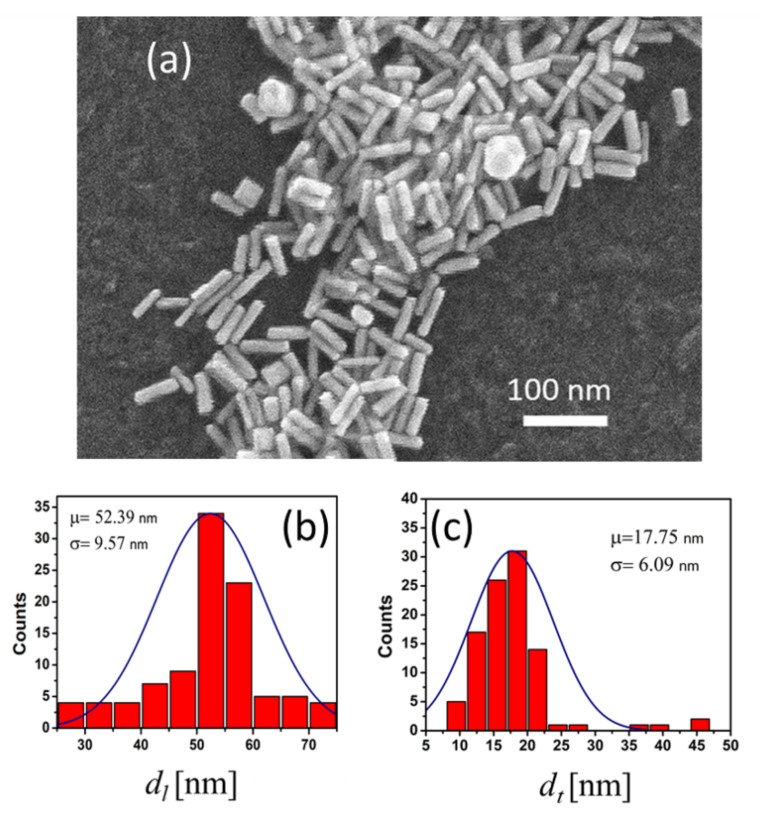
(**a**) SEM image of some of the gold nanorods synthesized. We plotted the histograms of the values of the major axis (**b**) and minor axis (**c**) of the fabricated nanoparticles. The solid line represents the Gaussian fitting where μ and σ are the mean and standard deviation of these distributions, respectively.

**Figure 2 nanomaterials-11-01696-f002:**
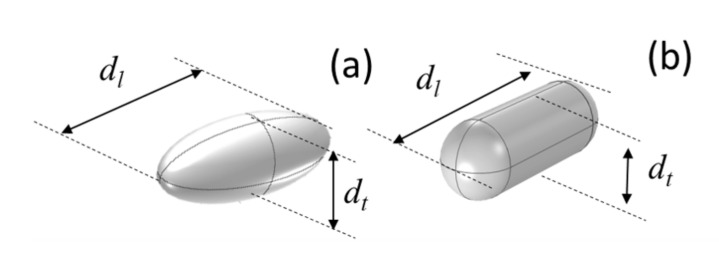
Geometrical models for the nanoparticles using (**a**) a prolate ellipsoid, and (**b**) a cylinder terminated with two hemispherical caps.

**Figure 3 nanomaterials-11-01696-f003:**
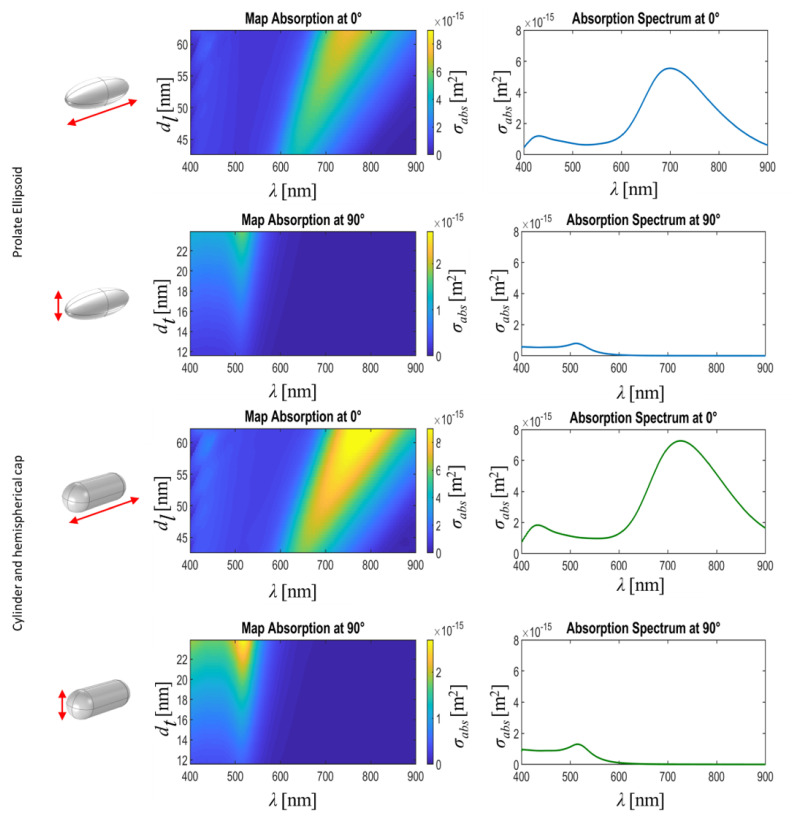
The spectral maps shows the calculated absorption cross section, $\sigma_{\mathrm{abs}}$, as a function of the wavelength, λ, and the most relevant dimension for each polarization: $d_{l}$ for θ=0°, and $d_{t}$ for θ=90°. The red arrow on the first column represents the orientation of the incoming electric field. The spectral maps and plots are obtained after a weighted average of the $\sigma_{\mathrm{abs}}$ considering the size distribution presented in Figure 1b,c.

**Figure 4 nanomaterials-11-01696-f004:**
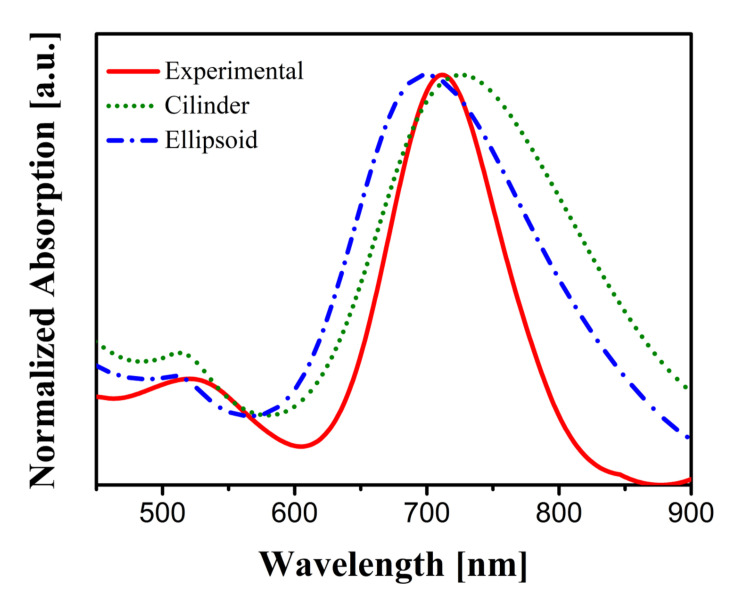
Comparison between experimental and simulatedspectral absorption. The numerical plots are the average of the two main polarizations at θ=0° and θ=90°.

**Figure 5 nanomaterials-11-01696-f005:**
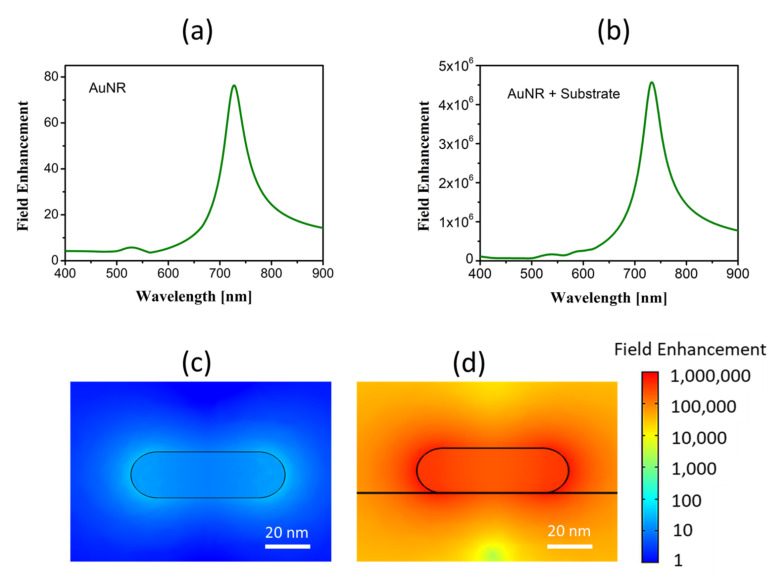
Maximum field enhancement as a function of wavelength for the case of (**a**) the nanorods immersed in water and (**b**) deposited on a Si substrate. Spatial maps of the electric field in a plane perpendicular that contains the axis of the nanorod at λ=728 nm for the case of (**c**) the nanorods immersed in water and (**d**) deposited on a Si substrate. They are represented in $\log_{10}$ scale using the same range.

**Table 1 nanomaterials-11-01696-t001:** Resonant wavelengths for the two simulated geometries and observed experimentallly.

	Prolate Ellipsoid	Cylinder + Hemispheres	Experimental
$\lambda_{l}\left(\theta =0{}^{\circ}\right)$ [nm]	698	720	712
$\lambda_{t}\left(\theta =90{}^{\circ}\right)$ [nm]	509	512	521

**Table 2 nanomaterials-11-01696-t002:** Correlation coefficient, *c*, between the experimental and the simulated geometries (see Equation (Equation 1)).

	Prolate Ellipsoid	Cylinder + Hemispheres
Longitudinal, θ=0°	0.9165	0.9799
Transversal, θ=90°	0.8587	0.8647

## Data Availability

The data obtained in this contribution can be accessed at https://figshare.com/s/7874575392cd028b932b.

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
