# Peer review of "Computational and Experimental Analysis of Gold Nanorods in Terms of Their Morphology: Spectral Absorption and Local Field Enhancement"

_nanomaterials, 2021, doi:10.3390/nano11071696_

Round 1
Reviewer 1 Report
The paper Computational and experimental analysis of the optical response of gold nanorods by Nunez Leyva et al . analyses the effect of the shape of gold nanorods on the plasmonic effect .The topic is not new ,however it gives a very sound and complete explanation of the Raman signal amplification .Normally researchers claim a large amplification but they do not demonstrate it .This paper shows the amplification effect very clearly .
This paper will be useful for advances of the field.
Author Response
We really appreciate your opinion that encourages us to improve the quality and relevance of our research.

Reviewer 2 Report
The paper is quite interesting and complete. It would be nice in a near future read a paper in which SERS measurements with this kind of nanorods are reported.
Author Response
We really appreciate your opinion that encourages us to improve the quality and relevance of our research. As we did in a previous collaboration, we are working to have these gold nanorods in a biomedical experiment to reveal the response of some molecules of interest. We are also looking forward to extending this analysis in a real-world application of Raman spectroscopy involving biological samples.

Reviewer 3 Report
The paper under consideration is concerned with optical properties of gold nanorods. The subject of the paper corresponds to one of the topics of the journal “Nanomaterials”.
My major comments on the manuscript:
1. The Introduction is not well written. The state-of-the-art is incompletely presented, and the objective of the particular study is not clearly formulated in the Introduction.
In a brief discussion of biomedical applications of plasmonic gold nanoparticles, it is recommended to mention the following journal papers on this subject:
N.G. Khlebtsov and L.A. Dykhman, Optical properties and biomedical applications of plasmonic nanoparticles, J. Quant. Spectr. Radiat. Transf. 111 (1) (2010) 1–35.
L.A. Austin, M.A. Mackey, E.C. Dreaden, and M.A. El-Sayed, The optical, photothermal, and facile chemical properties of gold and silver nanoparticles in biodiagnostics, therapy, and drug delivery, Arch. Toxicol. 88 (2014) 1391–1417.
L.A. Dombrovsky, V. Timchenko, M. Jackson, and G.H. Yeoh, A combined transient thermal model for laser hyperthermia of tumors with embedded gold nanoshells, Int. J. of Heat and Mass Transfer, 2011, v. 54, n. 25-26, pp. 5459-5469.
Y.L. Hewakuruppu, L.A. Dombrovsky, C. Chen, V. Timchenko, X. Jiang, S. Baek, and R.A. Taylor, Plasmonic “pump-probe” method to study semi-transparent nanofluids, Appl. Optics 52 (2013) 6041–6050.
X. Gu, V. Timchenko, G.H. Yeoh, L. Dombrovsky, and R. Taylor, The effect of gold nanorods clustering on near-infrared radiation absorption, Applied Sciences 8 (7) (2018) 1132.
These papers should be included in the list of references.
2. The first sentence of the Abstract (“It is well known that the shape and size of nanoparticles determine their optical properties”) is too general and not quite correct. The optical properties of nanoparticles of soot and many other substances are well described by the Rayleigh theory. It means that absorption coefficient of a cloud of such particles does not depend on the particle size. The absorption coefficient is directly proportional to the volume fraction of particles.
The following monographs are recommended for more details:
C.F. Bohren and D.R. Huffman, Absorption and Scattering of Light by Small Particles, Wiley, New York, 1983.
L.A. Dombrovsky and D. Baillis, Thermal Radiation in Disperse Systems: An Engineering Approach, Begell House, New York, 2010.
The references to these books are recommended also to avoid the wrong statements like “single-scattering solution of Maxwell’s equations” and also such a strange term as “Mie–Gans theory”. Note that this term is not used by researchers working in the field of radiation scattering by particles.
3. The references to papers [40–42] are not representative in the discussion of the discrete dipole approximation. As a minimum, it is recommended to refer to the following review paper:
M.A. Yurkin and A.G. Hoekstra, The discrete dipole approximation: An overview and recent developments, J. Quant. Radiat. Transf. 106 (2007) 558–589.
4. The technical English of the manuscript should be improved. Some examples: “experimental measurements”, “a complex or sophisticated measurement set-up”, “optical radiation”, etc. Please do not use the term “fitting” while talking about good agreement between the computational results and experimental data.
5. As I understand, there are no methodological novelties in the study reported by the authors. However, the particular results on the effect of the particle shape on the normalized absorption may be interesting for potential readers of the journal.
6. The image in Fig. 1a is not sharp and should be improved.
The paper can be recommended for publication after revisions according to the above comments.
